# Human Robot Hand Interaction with Plastic Deformation Control

**Kenichi Murakami** [1,*] , **Koki Ishimoto** [2], **Taku Senoo** [3] **and Masatoshi Ishikawa** [1]

[1] Information Technology Center, The University of Tokyo, 7-3-1 Hongo, Bunkyo-ku, Tokyo 113-8656, Japan; ishikawa@ishikawa-vision.org

[2] Department of Information Physics and Computing, Graduate School of Information Science and Technology, The University of Tokyo, 7-3-1 Hongo, Bunkyo-ku, Tokyo 113-8656, Japan; ishimoto@star.rcast.u-tokyo.ac.jp

[3] Electrical, Systems, and Control Engineering Program, Graduate School of Advanced Science and Engineering, Hiroshima University, 1-4-1 Kagamiyama, Higashi-Hiroshima City, Hiroshima 739-8527, Japan; taku-senoo@hiroshima-u.ac.jp

\* Correspondence: murakami@ishikawa-vision.org; Tel.: +81-3-5841-6938

**Abstract:** In recent years, force control has become more important due to the physical interaction of robots with humans and applications of robots to complex environments. Impedance control is widely used in force control; however, it cannot reproduce the behavior of plastic deformation because it returns to the initial position when the force is removed, similar to elastic deformation. On the other hand, Senoo et al. have proposed plastic deformation control based on the Maxwell model. However, because plastic deformation control is model-based, it is subject to the modeling and parameter errors of the controlled system. A robot hand is relatively small and lightweight; because it uses a gearbox with a high reduction ratio for its joints, it is significantly affected by friction and tends to deviate strongly from the desired motion. Therefore, in this study, a method that is robust against modeling and parameter errors is proposed by feeding back the error from the desired trajectory with the inner position loop. Then, the effectiveness of the proposed method is shown through simulations and experiments using an actual robotic system.

**Keywords:** shock absorb; plastic deformation control; inner position loop; human robot interaction; robot hand

## 1. Introduction

Currently, the real-time application of robots is being investigated. In addition to recognizing a complex environment using vision, it is critical for a robot to control the force with respect to the environment and human contact.

Several studies regarding the control of force received from the environment have been conducted. Mechanical methods using devices are examples of achieving the gentleness of robots against the relative external forces [1,2]. Although these approaches have a sufficient response bandwidth, it is challenging to use them in actual robots considering the device size and hysteresis. Therefore, this study focuses on the software servo control method to achieve mechanical stiffness. In software servo control, the impedance control method is widely used, for which several different approaches have been reported. For example, including feedback from positions and velocities without requiring force sensors [3], compensation to achieve the transformation to linear dynamics [4], and improvement of robustness to reduce the adverse effects of disturbance and model errors [5].

Herein, material deformation is expressed by the combination of elastic and plastic deformations. Impedance control can reproduce elastic deformation because the endpoint returns to the equilibrium point when the external force is removed. In contrast, Senoo et al. proposed a method for deformation

control that achieves plastic deformation behavior [6]. It is based on the Maxwell model in which a spring and damper are connected in series. The advantage of plastic deformation control over impedance control is low repulsive force to the environment. In the case of the impedance control, a repulsive force that returns to its original position of zero displacement is always generated when the equilibrium position changes due to external force. On the other hand, in plastic deformation control, the reaction force to the environment can be suppressed because the damper element absorbs the restoring force of the spring element, as illustrated in the maxerll model. Therefore, the applications of plastic deformation control can be considered, such as human–robot cooperation with more consideration for the safety of human workers, shock absorption, manipulation without deforming flexible objects, and so on. There are studies that have achieved plastic deformation control with an actual spring and a software-controlled damping element [7], and switching between the plastic and elastic deformation responses by expressing the Maxwell model in an impedance control system [8]. Consequently, elastoplastic deformation was achieved, called the "Zener model", which is a combination of plastic and elastic deformation [9]. However, the deformation control is model-based and is significantly affected by the model parameter errors. Particularly for robot hands, the behavior of the friction term becomes dominant because its inertia is relatively small. Moreover, the verification of the model parameter errors and friction, excluding the initial study cited above, has been limited to simulations.

Therefore, this study aims to propose a robust control law for modeling and parameter errors in order to achieve plastic deformation control with an actual robot hand system. The feedback of the difference between the desired behavior obtained by numerical calculation of the deformation model and the actual behavior, which is called the "Inner position loop" [5,10], into the plastic deformation control to suppress the effects of the model parameter errors, especially friction, is proposed. In previous studies, function approximation methods [11,12], a backstepping method [13], sliding mode control [14], learning method [15], and other adaptive methods [16–18] have been proposed for managing model and parameter errors. In this study, the proposed method was adopted because the controlling target is a relatively small and lightweight high-speed robot hand and finger, which can provide a high-speed feedback with a high control rate and is capable of an instantaneous high torque output.

The content of this study is presented as follows: Plastic deformation control and the proposed method are presented in Section 2. Section 3 provides the actual robot system. To verify the proposed method, simulation results and experimental results are shown in Sections 4 and 5. This paper focuses on the analysis of the proposed method by adding simulations, from the previous paper [19]. Finally, Section 6 provides the conclusion of the study.

## 2. Plastic Deformation Control

The Voigt model and the Maxwell model are known as the fundamental models used to describe the behavior of viscoelastic materials [20]. More applied models such as the standard linear solid model (SLS or Zener model) and the four element model have a structure that includes these elements. The Voigt model, in which a spring and a damper are connected in parallel as shown in Table 1, is suitable for illustrating elastic deformation in which the displacement returns to zero when an external force is removed. The impedance control method corresponds to this model.

In contrast, the Maxwell model, in which a spring and a damper are connected in series, is suitable for illustrating plastic deformation, where the displacement does not return to zero despite the removal of an external force. Considering this model, Senoo et al. proposed the plastic deformation control [6]. In this control method, the shifts in the position and posture attributable to the external force are regarded as plastic deformation.

**Table 1.** Comparison between the fundamental deformation models.

| Model | Voigt Model | Maxwell Model |
|---|---|---|
| Deformation type | Elastic deformation | Plastic deformation |
| Connection configuration | Parallel | Series |
| Corresponding control law | Impedance control | Plastic deformation control |
| Diagram |  |  |

### 2.1. Nominal Control Law

In this study, the interaction with a serial-link robot finger is considered. Then, the dynamics of the finger with the joint variable $q$ are computed in the joint space as follows:

$$M_q \ddot{q} + h = \tau + J^T F \tag{1}$$

Here, $M_q$ for the inertia matrix of the robotic finger, $h$ for the sum of the Coriolis force, centrifugal force, gravity, and viscous friction; $J$ for the Jacobian matrix, and $F$ for the external applied force.

Defining $r$ as the endpoint position, it is derived from the relationship between the endpoint position and the joint angle as follows:

$$\dot{r} = J\dot{q}, \tag{2}$$

$$\ddot{r} = \dot{J}\dot{q} + J\ddot{q}. \tag{3}$$

Then, Equation (1) is transformed by substituting Equation (3), and multiplying by $J^{+T}$ from the left gives

$$\left( J^{+T} M_q J^+ \right) \ddot{r} + J^{+T} h - J^{+T} M_q J^+ \dot{J}\dot{q} = J^{+T}\tau + F. \tag{4}$$

Here, $J^+$ for the pseudo-inverse matrix of the Jacobian.

On the other hand, the equation of motion of the Maxwell model can be obtained from its diagram in Table 1.

$$M\ddot{x} = F - K\left(x - p\right) \tag{5}$$

$$K\left(x - p\right) - C\dot{p} = 0 \tag{6}$$

Here, $M$ stands for mass, $K$ for spring constant, $C$ for viscosity coefficient, $x$ for displacement of the mass and $p$ for displacement of the damper. From these equations, the equation of the motion can be expressed as follows:

$$M\ddot{x} + KC^{-1}M\dot{x} + Kx = F + KC^{-1}\int F dt. \tag{7}$$

By designating the desired inertia, stiffness, and damping matrix as $M_d$, $K_d$, and $C_d$, respectively, the target acceleration is derived from Equation (7) as follows:

$$\ddot{r}_d = -M_d^{-1}\left( K_d C_d^{-1} M_d \dot{r}_e + K_d r_e - F - K_d C_d^{-1}\int F dt \right). \tag{8}$$

Here, $r_d$, and $r_e$ are the desired and current positions of the endpoint, respectively.

Then, the control law of the plastic deformation control is derived by substituting $\ddot{r}_d$ in Equation (8) for $\ddot{r}$ in Equation (4) as follows:

$$\tau = \tau_{NC} + \tau_{VE} + \tau_{IN} + \tau_{PL} \tag{9}$$

$$\text{where} \quad \tau_{NC} \equiv h - M_q J^+ \dot{J} \dot{q},$$

$$\tau_{VE} \equiv -E\left(K_d C_d^{-1} M_d \dot{r}_e + K_d r_e\right),$$

$$\tau_{IN} \equiv \left(E - J^T\right) F,$$

$$\tau_{PL} \equiv E K_d C_d^{-1} \int F dt,$$

$$E \equiv M_q J^+ M_d^{-1}.$$

Here, $\tau_{NC}$, $\tau_{VE}$, $\tau_{IN}$, and $\tau_{PL}$ indicate non-linear compensation, the viscoelastic setting, inertia adjustment, and the plastic expression, respectively.

### 2.2. Plastic Deformation Control with the Inner Position Loop

Figure 1 presents a block diagram of the plastic deformation control. The control flow first calculates the input acceleration $\alpha$ by using the deformation model based on the deformation equilibrium point $r_0$ and feedback information. Next, the torque input $\tau$ that determines the deformation model response is calculated. Then, a manipulator is controlled by applying this torque.

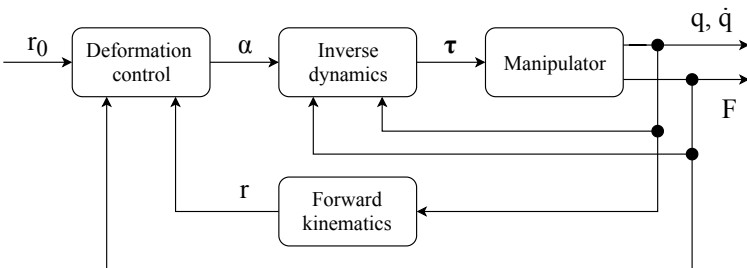

**Figure 1.** Block diagram of plastic deformation control.

However, it cannot present the desired behavior if there are modeling or parameter errors because this control law is model-based. Friction has a significant effect on the robot's behavior considering its hands because of its relatively small inertia. Moreover, friction is considerable, and estimation of friction is challenging because the reducer in each joint presents non-linear characteristics and hysteresis.

Therefore, the feedback of the difference between the current and desired positions by numerical calculation of the input acceleration $\alpha$ for robust control against model parameter errors and friction effects is proposed, as shown in Figure 2. This method is called the "Inner position loop (IPL)" [5,10]. The input acceleration $\alpha_c$ is calculated as follows:

$$\alpha_c = \ddot{r}_e + G_v\left(\dot{r}_c - \dot{r}_e\right) + G_p\left(r_c - r_e\right). \tag{10}$$

Here, $r_c$ is the desired position of the endpoint obtained by the numerical calculation of the deformation model, Equation (8), with the measured external force; $G_p$ and $G_v$ stand for proportional and derivative gains, respectively. This time, the Runge–Kutta fourth-order method is used to calculate $r_c$.

Next, considering Equation (3), the following equation is obtained.

$$\ddot{q} = J^+ \left(\alpha_c - \dot{J} \dot{q}\right) \tag{11}$$

Then, the proposed control law is derived from Equations (4), (8), (10) and (11) as follows:

$$\boldsymbol{\tau} = \boldsymbol{\tau}_{NC} + \boldsymbol{\tau}_{VE} + \boldsymbol{\tau}_{IN} + \boldsymbol{\tau}_{PL} \tag{12}$$

where
$$\boldsymbol{\tau}_{NC} \equiv \boldsymbol{h} + \boldsymbol{M_q}\boldsymbol{J}^{+}\left(\boldsymbol{G_v}\left(\dot{\boldsymbol{r}}_c - \dot{\boldsymbol{r}}_e\right) + \boldsymbol{G_p}\left(\boldsymbol{r}_c - \boldsymbol{r}_e\right) - \dot{\boldsymbol{J}}\dot{\boldsymbol{q}}\right),$$

$$\boldsymbol{\tau}_{VE} \equiv -\boldsymbol{E}\left(\boldsymbol{K_d}\boldsymbol{C_d}^{-1}\boldsymbol{M_d}\dot{\boldsymbol{r}}_e + \boldsymbol{K_d}\boldsymbol{r}_e\right),$$

$$\boldsymbol{\tau}_{IN} \equiv \left(\boldsymbol{E} - \boldsymbol{J}^{T}\right)\boldsymbol{F},$$

$$\boldsymbol{\tau}_{PL} \equiv \boldsymbol{E}\boldsymbol{K_d}\boldsymbol{C_d}^{-1}\int \boldsymbol{F}dt,$$

$$\boldsymbol{E} \equiv \boldsymbol{M_q}\boldsymbol{J}^{+}\boldsymbol{M_d}^{-1}.$$

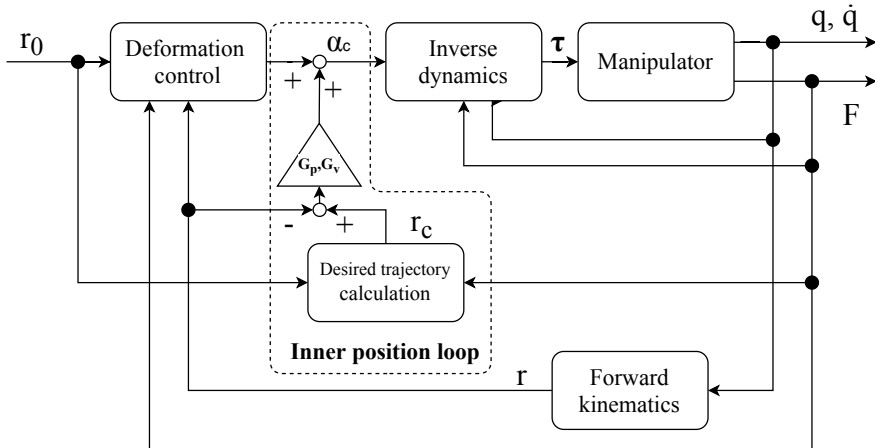

**Figure 2.** Block diagram of the plastic deformation control with inner position loop.

## 3. System Configuration

The system configuration is shown in Figure 3. This system consists of a high-speed robotic finger with a force sensor and a real-time control system.

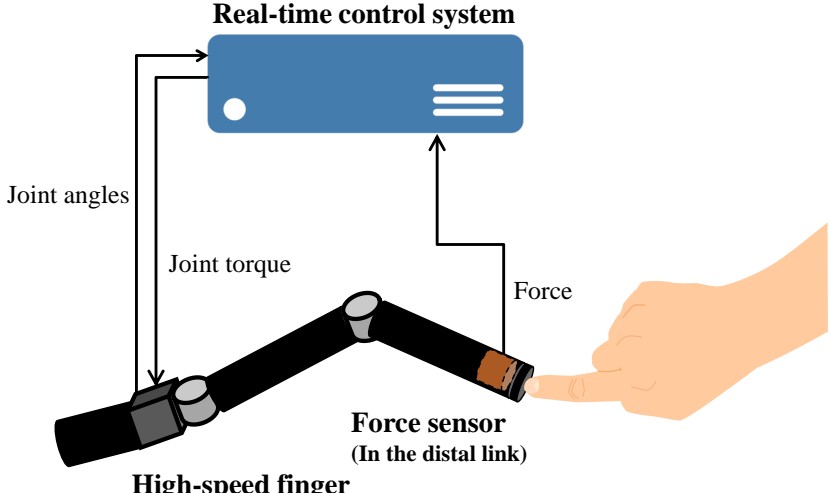

**Figure 3.** System configuration.

### 3.1. High-Speed Finger with a Force Sensor

The high-speed finger in the high-speed hand that was developed for dynamic manipulation [21] and used in high-speed throwing [22] is chosen and a force sensor is installed in its distal link. This finger has 2 joints and moves in a horizontal plane, as shown in Figure 4. $L_1$ and $L_2$ in Figure 4b are 65 mm and 60 mm respectively. A small harmonic drive gear® and a high-power mini actuator are installed in the finger link. The design of this actuator, which is based on the concept that the maximum power output, rather than rated power output, should be improved. Table 2 presents the specification of actuators used in this finger. This finger can close its joints at 180 deg per 0.1 s; its maximum rotational speed is 300 rpm, and the maximum output force is 12 N.

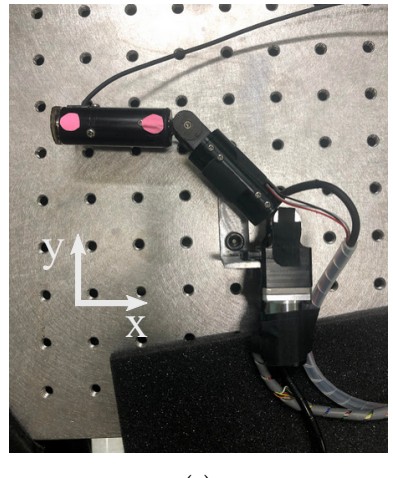

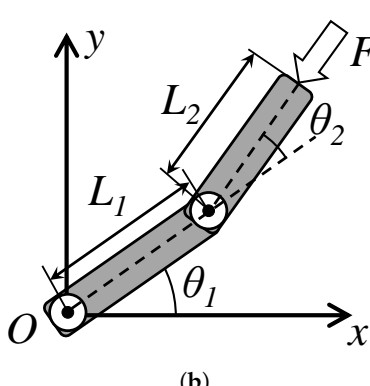

(a)                                              (b)

**Figure 4.** High-speed finger: (**a**) Overview of the finger. (**b**) Its kinematic model.

**Table 2.** Specification of actuators used in the high-speed finger.

|                      | Joint 1 | Joint 2 |
|----------------------|---------|---------|
| Reduction rate       | 50      | 50      |
| Maximum torque [Nm]  | 0.9     | 0.21    |
| Maximum speed [rpm]  | 200     | 200     |

TFSA12-10 manufactured by Japan liniax is used as the force sensor. It is a relatively small and light-weighted 6-axis force torque sensor for which the response frequency is approximately 5 kHz and the measuring range is 20 N in the axial direction, 10 N in the radial direction. The distal link of the finger is modified from a high-speed hand setup, and a force sensor in the distal link is installed to match the endpoint of the finger and sensor surface, as shown in Figure 3.

### 3.2. Real-Time Control System

The robot finger is controlled by a real-time controller (dSPACE, CPU: PowerPC 750GX (1 GHz)). It reads the force values from the force sensor and controls the finger by using the proposed control. These operations are performed every 1 ms.

## 4. Simulation

The movement of the two-joint finger was simulated by using the proposed control. The Open Dynamics Engine (ODE) was used for simulation. Here, the same mechanical parameters as those of the actual high-speed finger were used. The control rate was also set to 1 kHz according to the actual system.

### 4.1. Simulation with Fiction Error

First, the case regarding viscous friction error was simulated by applying a constant force. A $F_x = F_y = 0.2\,\text{N}$ force for $0.5\,\text{s}$ was applied at the endpoint of the finger. The viscous friction coefficient was set to $(0.05, 0.05)$ in the model, and $-0.01$ was added as the error in the control. $\boldsymbol{M_d}$, $\boldsymbol{K_d}$, and $\boldsymbol{C_d}$ were set to $diag\,(0.5, 0.5)$, $(100, 100)$, and $(2.0, 2.0)$, respectively. The gain parameters were set to $\boldsymbol{G_p} = (10{,}000, 10{,}000)$, and $\boldsymbol{G_v} = (1000, 1000)$.

The simulated results are shown as the trajectory of the endpoint of the finger in Figure 5. As shown in the figure, by using the proposed control, the endpoint of the finger passes close to the desired trajectory; however, the trajectory significantly deviates when nominal control (plastic deformation control without IPL ) is used.

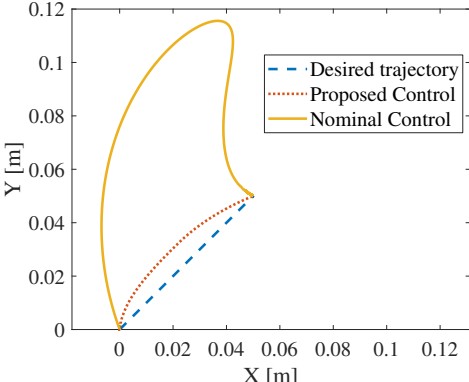

**Figure 5.** Trajectory of displacements of the endpoint in the simulation with error added to the friction coefficient.

This may be caused by a delay of movement due to a lack of input torque caused by a friction coefficient error, as shown in Figure 6.

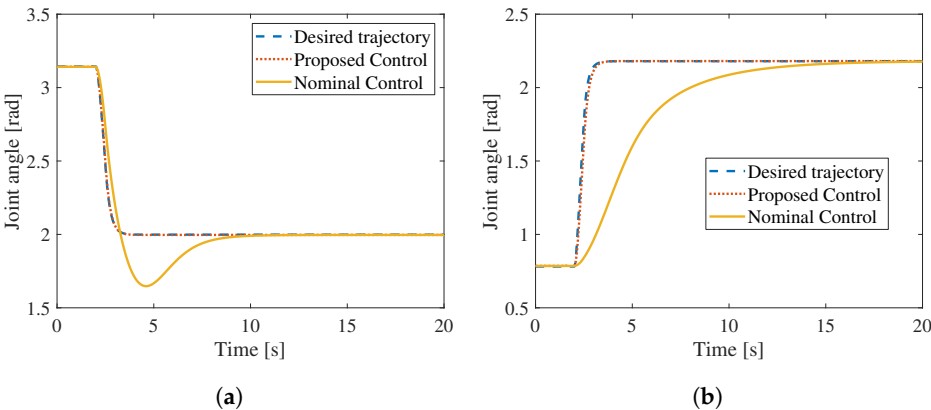

| (a) | (b) |

**Figure 6.** Trajectory of the joint angles in the simulation with error added to the friction coefficient: (**a**) Joint 1. (**b**) Joint 2.

Then, it is considered that the deviation of the trajectory of the end point is mainly due to the significant delay of the second joint because the influence of the friction error was relatively large due to the inertia of the distal link being smaller than that of the root link. On the contrary, in the case of the proposed method, it is considered that the trajectory of the endpoint was closer to the desired trajectory by compensating the input torque with the feedback of error from the desired trajectory. In addition, these figures present that the trajectories converge at the same point, regardless of the trajectories. This result is supported by the fact that the convergence distance can be calculated as $r_e(\infty) = C_d^{-1} \int F dt$ by substituting the static condition of $\dot{r}_e = \ddot{r}_e = F(\infty) = 0$ into Equation (7).

In order to verify this, the movement was simulated by changing the modeling parameters of the plastic deformation control. Simulations were performed under two conditions: one is the condition where $M_d$ ($= diag\,(2.0, 2.0)$) and $K_d$ ($= (1000, 1000)$) were changed without changing $C_d$ and, another is the condition where $M_d$ ($= diag\,(2.0, 2.0)$), $K_d$ ($= (1000, 1000)$) and $C_d$ ($= (4.0, 4.0)$) were changed.

Figure 7 presents the trajectories of the end point. When $C_d$ is the same, they converge to the same point despite $M_d$ and $K_d$ being changed and their trajectories are different. When $C_d$ is doubled, it converges to $X = Y = 0.025\,\mathrm{m}$ which is half of $X = Y = 0.05\,\mathrm{m}$ in other cases. Furthermore, because the applied force is $F_x = F_y = 0.2\,\mathrm{N}$, the convergence point is apparently in the direction of the force.

Next, the variation of force over time was simulated. First, the case where the force is being applied by separating in the $X$ and $Y$ directions was simulated. The force was $0.2\,\mathrm{N}$ for $0.5\,\mathrm{s}$ in the order of $Y$ and $X$ as shown in Figure 8a. As shown in Figure 8b, the trajectories are different, but they converge to the same point as when the forces are applied in both axes simultaneously (SIMUL.).

Second, the application of force while changing the circular direction was simulated as shown in Figure 9a. Considering Figure 9b, the desired trajectory is curved and the trajectory with the proposed control follows it.

From these simulation results, it was experimentally validated that the proposed method is robust against the error of the viscous friction coefficient, regardless of the model parameters and the input external force.

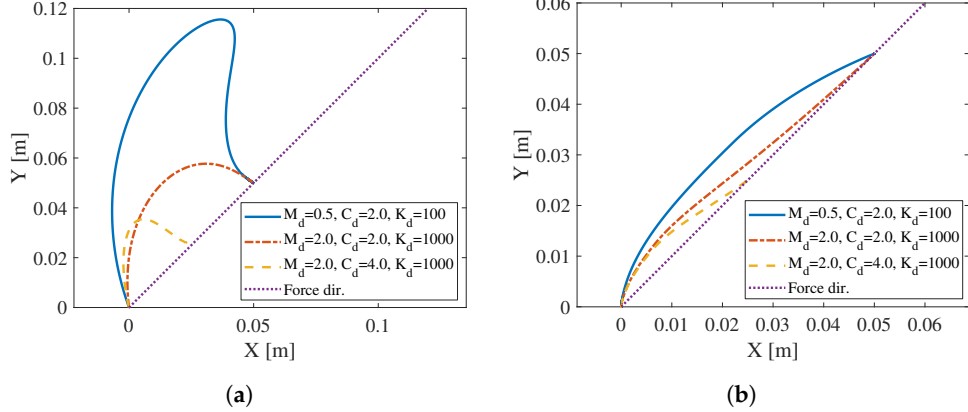

**Figure 7.** Trajectory of displacements of the endpoint with varying parameters: (**a**) With nominal control. (**b**) With proposed control.

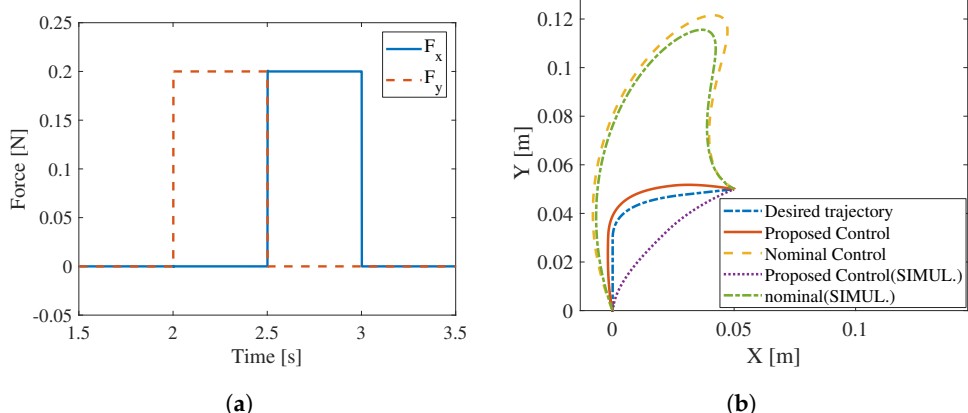

**Figure 8.** Simulation with applying force by separating in the $X$ and $Y$ directions. (**a**) Applied force. (**b**) Trajectories of end point.

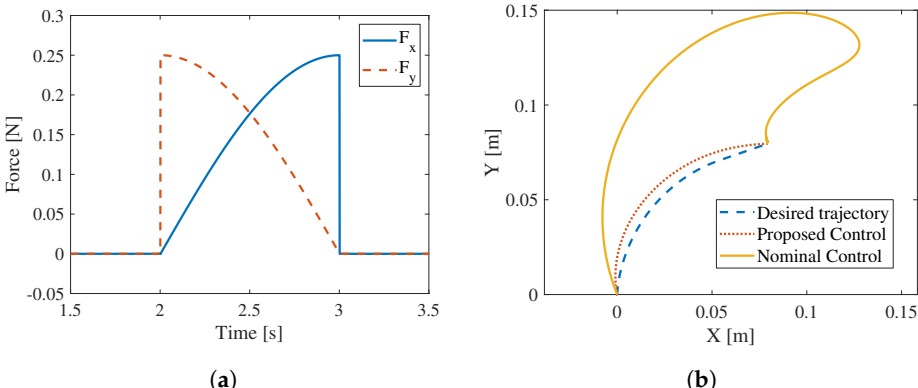

**Figure 9.** Simulation with applying force while changing the direction to circular. (**a**) Applied force. (**b**) Trajectories of end point.

*4.2. Simulation by Varying the Parameter with Error*

Next, the effect of parameter errors was simulated. The movements of the first case were simulated by changing the model parameter that adds the error. At that point, 20% errors were added to the position of the center of gravity, mass, link length, and link radius in the control similar to the viscous friction coefficient error in the first case. Figure 10 presents the trajectory of displacements of the endpoint. As shown in these figures, the deviation is the largest when there is an error in the viscous friction coefficient. For example, the case with the link length error where the deviation is the second-largest with the nominal control is smaller than that of the case with the viscous friction error with the proposed control. Moreover, in cases other than with friction error, the trajectories of the endpoint are approximately the same as the desired trajectory obtained by using the proposed control.

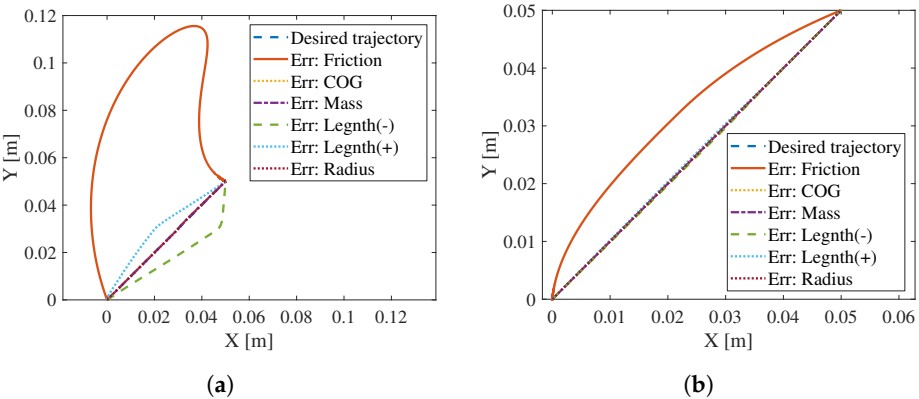

**Figure 10.** Trajectory of displacements of the endpoint in the simulation by varying the parameter with error: The labels in the legend starting with "Err: " show the parameter with 20% error added. (**a**) With nominal control. (**b**) With proposed control.

The influence of other parameter errors was considered to be smaller than that of the friction error because the simulation conditions were set according to the high-speed robot finger which is small and lightweight. In the actual robot system, a reducer with the actual high reduction ratio is used for each joint of the finger, and it is difficult to model their friction. Therefore, it is more reasonable to compensate for the error by feeding back the difference from the desired trajectory with the proposed control rather than eliminating the model error of friction.

## 5. Experiment

In order to verify the effectiveness of the proposed control, the following two types of experiments were conducted: applying a virtual constant force to the endpoint of the robotic finger and pushing the endpoint with a human finger. The results were compared with those of nominal control.

### 5.1. Applying Virtual Constant Force

First, a virtual constant force was applied to the endpoint of the robotic finger. A $F_x = F_y = 0.2\,\text{N}$ force was applied for 0.5 s at the fingertip virtually by overwriting the sensor input in the controller. Its initial posture was set to make the distal link parallel to the $x$ axis, as shown in Figure 4a. In this experiment, the control parameters were selected as follows: The viscous friction coefficient, for which the actual value was not measured, was set to $(0.02, 0.02)$. $\boldsymbol{M_d}$, $\boldsymbol{K_d}$, and $\boldsymbol{C_d}$ were set to $diag\,(0.5, 0.5)$, $(100, 100)$, and $(2.0, 2.0)$, respectively, similar to the simulation. The gain parameter was set to the same value in the $x$ and $y$ directions $\boldsymbol{G_p} = (4000, 4000)$, and $\boldsymbol{G_v} = (520, 520)$, respectively. These parameters were determined heuristically.

Figures 11 and 12 present successive images of the experimental results and the trajectory of its endpoint and joint angles, respectively. As shown in the figures, the fingertip moved in the direction of the desired trajectory using the proposed control, whereas the fingertip slightly moved using the nominal control, which indicates that the generated torque was smaller than that of the static friction using the nominal control, while it was compensated for in the proposed control. However, the trajectory significantly deviated from the desired trajectory with the proposed control. We consider this is because while the gains of IPL were set in the task space (XY), the actuators used in each joint were different as shown in Table 2. Therefore, the different joint frictions caused the difference in the start time of moving each joint. However, this can be solved by calculating the desired trajectory of the finger and setting the gains in the joint space rather than in the task space.

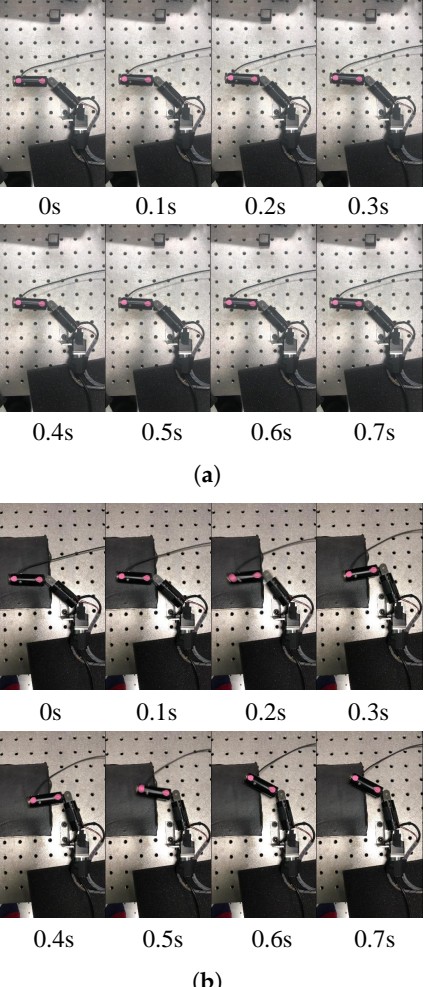

**Figure 11.** Successive images of the experiment: virtually applying $F_x = F_y = 0.2\,\text{N}$ for 0.5 s. (**a**) With nominal control. (**b**) With proposed control.

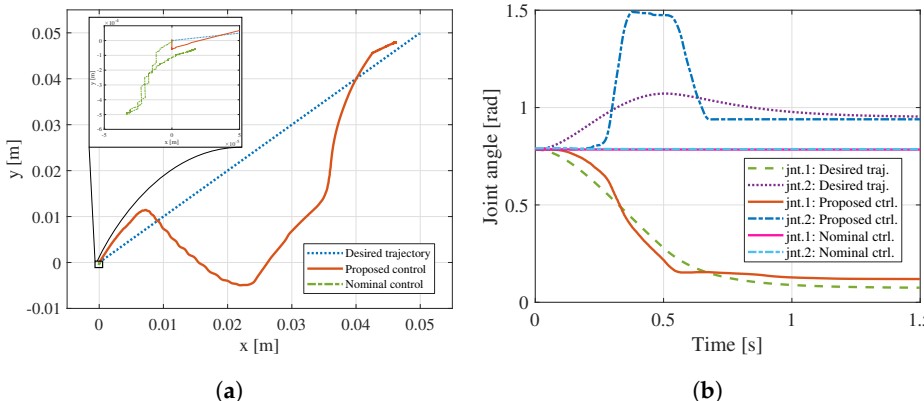

**Figure 12.** Response of the finger by applying a virtual constant force: (**a**) Displacement of the end point. (**b**) Trajectories of the joint angle.

## 5.2. Pushing the Fingertip with Human Finger

Next, the fingertip was experimentally pushed using a human finger. Here, the human pushed the endpoint of the finger in the $x$ direction. The initial posture of the finger and control parameters were the same as those of the previous experiment.

Figures 13 and 14 present successive images of the experiment and trajectories of the endpoint, respectively. As shown in these figures, using the proposed control, the fingertip smoothly moved while keeping contact with the human finger and achieved plastic behavior, although the robot finger had limited backdrive capability with stiff joints. In contrast, in the case of the nominal control, the fingertip vibrated significantly.

First, regarding nominal control, the finger was considered to vibrate due to the following: A large force was applied because the fingertip could not move due to static friction, similar to the previous experiment. Then, significant joint torque is generated because of the large applied force. This significant joint torque caused the finger to suddenly move when it exceeded the static friction. Furthermore, the finger passed the singular configuration by the effect of inertial force, and the Jacobian calculation was not performed correctly.

In contrast, with the proposed control, the fingertip moved along the desired trajectory in the beginning, and the measured force was not significant, as shown in Figure 15. However, the displacement in the $y$ direction gradually deviated from the desired trajectory when the displacement in the $x$ direction reached 0.025 m. This is because Joint 2 reached the mechanical limit, as shown in Figure 16, whereas the model used for calculating the desired trajectory does not have limitations in the joint angles. Although a joint limit can be set in the model, the applicable range becomes narrow. Therefore, it is better to improve the hardware to extend the movable range of the joint.

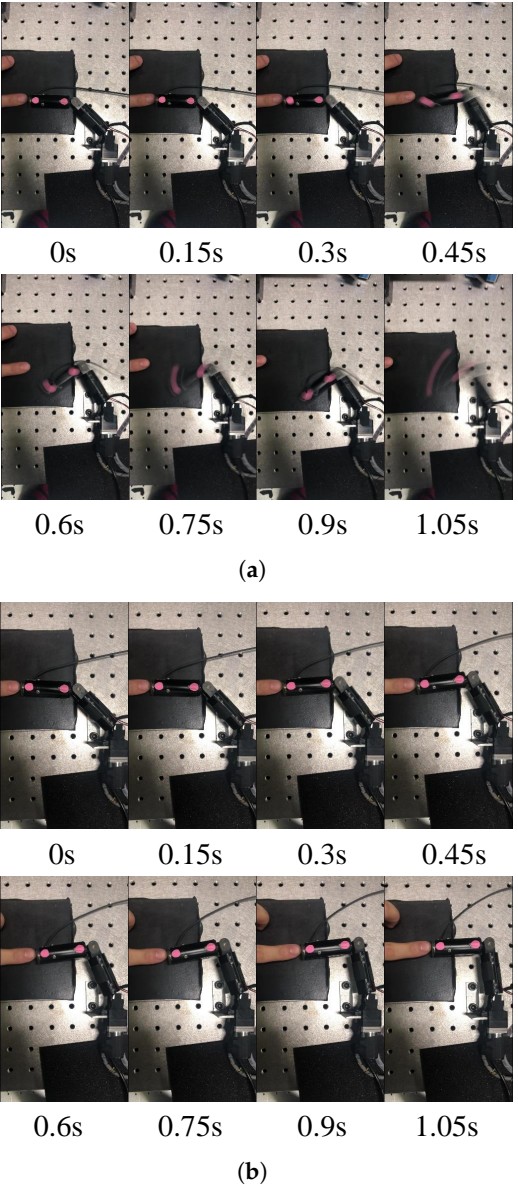

**Figure 13.** Successive images of the experiment: pushing with a human finger. (**a**) With nominal control. (**b**) With proposed control.

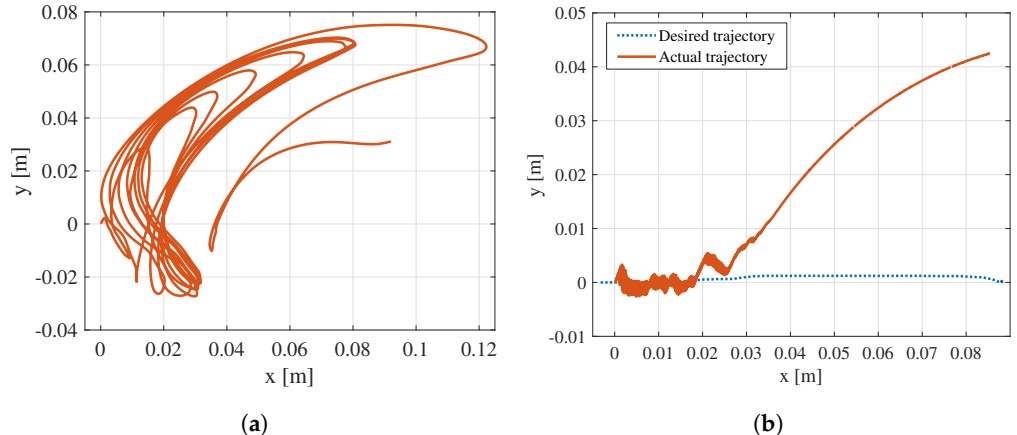

**Figure 14.** Displacement of the endpoint of the finger. (**a**) With nominal control. (**b**) With proposed control.

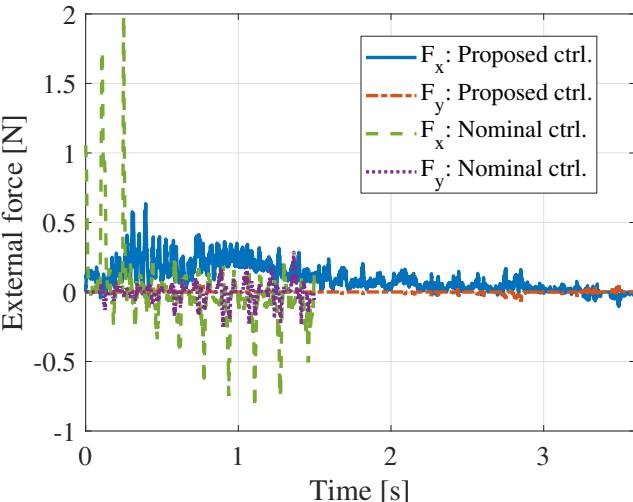

**Figure 15.** Measured external force on the fingertip.

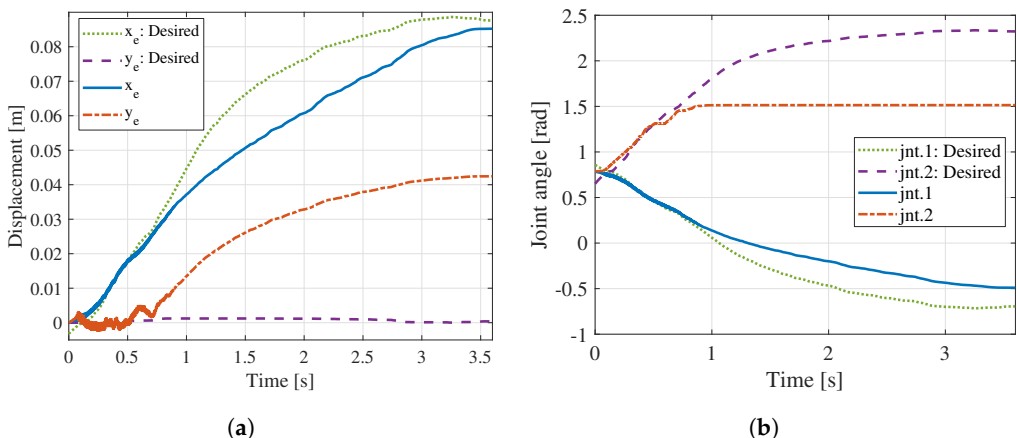

(**a**)          (**b**)

**Figure 16.** Response of the finger with the proposed control: (**a**) Trajectories of the endpoint. (**b**) Trajectories of joint angle.

## 6. Conclusions

In this study, in order to achieve plastic deformation control with an actual robot hand system the robust control law for modeling and parameter errors was proposed. Plastic deformation control for robots which have modeling and parameter error is achieved by feeding back the difference between the desired behavior obtained by the numerical calculation of the deformation model and the actual behavior. The effectiveness of the proposed control and robustness against the model parameter errors were validated experimentally by simulations and experiments using an actual system.

The proposed control can be improved in the future to move closer to the desired trajectory by feeding back errors in the joint space rather than in the task space and adjusting the gains for each joint. Furthermore, it is necessary to consider dealing with a singular configuration in order not to limit the range of movement. As a future study, robotic catching will be considered. Regarding robotic catching, impact absorption at the moment of contact is as critical as that after contact. Therefore, the proposed control will be combined with motion generation that minimizes the difference between the velocity of the endpoint and target at the time of contact, or a method that applies a virtual force according to the distance between the target and the endpoint to achieve robotic catching considering impact absorption.

**Author Contributions:** Conceptualization, T.S. and M.I.; Data curation, K.I., K.M. and T.S.; Funding acquisition, T.S. and M.I.; Investigation, K.I., K.M. and T.S.; Methodology, T.S. and K.I.; Project administration, T.S. and M.I.; Resources, K.M., T.S. and K.I.; Software, T.S., K.M. and K.I.; Supervision, T.S. and M.I.; Validation, K.I. and K.M.; Visualization, K.M. and T.S.; Writing—original draft, K.M.; Writing—review & editing, K.M., T.S., M.I. and K.I. All authors have read and agreed to the published version of the manuscript.

**Funding:** This research received no external funding.

**Conflicts of Interest:** The authors declare no conflict of interest.

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
