# Peer review of "Human Robot Hand Interaction with Plastic Deformation Control"

_robotics, doi:10.3390/robotics9030073_

Round 1

Reviewer 1 Report

The paper adds a set of experiments to a theoretical account that was published by the same group during the last few years (cited as Refs. [6, 7, 8]). It should also be considered that the control-theoretical approach was superseded meanwhile by a recent paper on the Zener model also by the same group. Other literature on these approaches goes back several decades, therefore similarities with earlier work are unavoidable.

The comparison in the second experiment appears to be biased against the "nominal control" case: Vibrations should be avoidable by considering bounds of the gain. Also, simple derivative control would be useful here as well. Also, if a singular configuration are reachable within a control system, precautions need to be taken, and it is not clear how the proposed control method would react when encountering a singular configuration. If this is considered, then the confounds in the explanation of the behaviour are more easily resolvable.

The modelling of the friction in the simulation seems more like a load applied to the finger, because it lacks a velocity-dependent component, Also in effect it seems quite strong. Wouldn't it make more sense to use a different friction model or to test the realism of this friction model in the robot experiment where actual friction can be added, rather than to carry over the suboptimal friction model from the simulation?

A few minor points should also be corrected:

Although "plastic deformation" appears as a the target problem, I would appreciate if other application cases where the same principle appears are mentioned and discussed as well.

Can you include an original reference for the Voigt and Maxwell models? The given Ref. [6] does not provide much more explanation about the origin of the models than included here.

9: "and does not perform the desired motion." Perhaps a bit to strict because this depends on the type of perturbation: What about (e.g.): "tends to deviate [strongly] from the desired motion"?

34: Don't use the equality sign here. Instead alpha should be properly defined.

Fig. 10: Try to improve the key in the pictures: Although it is not hard to guess, a curve that shows a trajectory should not be labelled, e.g., as "friction coefficient". Please, try to make more clear that you mean trajectory under variation of the friction coefficient etc.

Insert spaces before brackets and after dots!

Reviewer 2 Report

A  good research on the plastic deformation control of serial robot hand, necessary simulation and experiments have been conducted to verify the proposde method. Some suggestions are as follows.

1) It is too short for sec 3.2; For example the software development should be introduced;

2) Expessions including figures indication and euqations should be improved.

3) There exist a lot of informal expressions on language, mathematical equations, the authors are advised to follow the manuscript format carefully. It is better to inivite native speaker to polish the paper.

4) Line75-79, the identity symbol is incorrect;

5) The triangular block in figure 2 should be indicated;

6) Table 2 is missing a line;

7) In Sec 4, a picture of simualtion process should be illustrated;

8) (Fx, Fy) = (0.2, 0.2) is not a formal expression;

9) Line 130 "simulated results".

Reviewer 3 Report

The paper is about plastic deformation control of a robotic finger. The authors propose a control algorithm based on the Maxwell model with position feedback. The results are tested with simulations and experiments with real hardware. The results are interesting, however, the presentation and the analysis should be improved.

My comments are:

  1. The authors should emphasize the advantages of plastic control in the introduction. In the current form, it is hard to decide why plastic control is required. Also, there should be a more detailed comparison between plastic control and impedance control. 
  2. The derivations should be made more clear. In (2) and (3), r is not defined. Moreover, these equations give the relationships between end-effector velocities and accelerations and joint angle velocities and acceleration, although the text preceding the formulas states that "Using the Jacobian matrix, the relationship between the endpoint position and the joint angles is expressed as follows:", which is not correct in this form.
  3. Before (4), the authors should add "by substituting (3), and multiplying by J^+T from the left gives " to make the derivation more clear.
  4. How is (5) acquired from Table 1? What are the parameters? Please give the details of the derivation or give references (since the whole control is based on this formula).
  5. Please correct the typesetting for the dots over r_e and r_d, they are not over r, but the whole characters with the subscript included.
  6. The stability and robustness analysis of the proposed algorithm is completely missing. Using an algorithm in robot control without proving its stability is problematic. Also, the authors mention that their algorithm is "robust against modeling and parameter errors", however, this claim is not proved in the paper.
  7. In Table 2, what does weight refer to? The weight of the joint? Is it the motor and gearbox? Or is it the weight of the preceding or proceeding segment?
  8. Using subsections composed of one sentence like Subsection 3.2. should be avoided.
  9. What is the desired trajectory? Does it contain plastic deformation or not? It is not clear from the text. If it does not contain the plastic deformation, then the displacement from the desired trajectory should not be a problem, since this is caused by the external force. Driving the trajectory back using position feedback seems to be working against the plastic deformation. Please address this issue and clarify this in the paper.
  10. In line 142, the static conditions should also include F(\infty) = 0.
  11. What does steady stand for in the labels in Figure 8 (b)? 
  12. I think the results should be discussed in more detail with regards to the application which makes the usage of plastic deformation necessary, moreover, a comparison should be carried out with impedance control to prove the necessity of plastic deformation.
